# GM-Ledger: Blockchain-Based Certificate Authentication for International Food Trade

**DOI:** 10.3390/foods12213914

**Published:** 2023-10-25

**Authors:** William George, Tareq Al-Ansari

**Affiliations:** College of Science and Engineering, Hamad bin Khalifa University, Doha P.O. Box 34110, Qatar; williamgeorge.qatar@gmail.com

**Keywords:** blockchain technology, food certifications, digitized food logistics, smart contracts, blockchain architecture, food imports–exports, tokenization, distributed ledger technology

## Abstract

Maritime transportation plays a critical role for many Arab countries and their food security and has evolved into a complex system that involves a plethora of supply chain stakeholders spread around the globe. This inherent complexity brings huge security challenges, including cargo loss and high burdens in cargo document inspection. The emerging blockchain technology provides a promising tool to build a unified maritime cargo tracking system critical for cargo security. This is because blockchains are a tamper-proof distributed ledger technology that can store and track data in a secure and transparent manner. Using the State of Qatar as a case study, this research introduces the Global Maritime Ledger (GM-Ledger), which will aid authorities in verifying, signing and transacting food certificates in an efficient manner. The methodology of this research includes reviewing past publications, identifying the requirements of various players in the Qatari food import–export industry and then creating a smart contract framework that will efficiently manage the work with necessary human intervention as and when required. The result of this work is the formation of a solid framework that can be employed in future works. This work realized that employing web3 solutions for the food import sector is highly viable and that with the right social, economic and policy reforms, it is possible to transform the entire food system to bear healthy transparency and power balance in global supply chains. In conclusion, this study argues that BCT has the ability to assist the government and other players to minimize fraud and maximize food supply chain stakeholder participation.

## 1. Introduction

Food certifications are an important part of the international food trade and are necessary to guarantee the quality, safety and sustainability of food products as they move through the supply chain. These certifications, which give customers and businesses assurance in the goods they purchase, help the global food trade function smoothly. The role of the national government in food security is indispensable to the healthy living and prosperity of a nation. This role is understated and often considered irrelevant to investing in technology infrastructure. Every neoteric food policy presupposes a robust and versatile technology infrastructure. This facility is then utilized in unveiling, surveilling and evaluating the efficiency and sufficiency of the policy. The Nordic countries (Denmark, Finland, Norway and Sweden) [1] and some European nations (Luxembourg, Germany and Estonia) have governments that are recognized for their extensive adoption of technology in national food security governance. For example, as part of its strategic relationship with the World Food Program, the Luxembourg Government seeks to bolster progress toward Zero Hunger by adopting a common UN blockchain system, which will steer programs that engage the effectiveness of cash-based interventions (CBIs) and leverage food traceability [2]. Another illustrative instance is the visionary roadmap of the Ministry of Rural Affairs and the Ministry of Enterprise and Innovation in Sweden that aims to invest in technology to foster food system collaboration, empowering farmers and ameliorating knowledge management within different stakeholders of the food system [3,4].

As shown in Figure 1, the digitization of the global food trade is playing a bigger role in the development of world economies and comes with many advantages. The process can be made more productive, swifter and transparent while also improving the traceability and security of food products by utilizing digital platforms and technologies, like the blockchain. From farmers and producers to traders and consumers, this will be advantageous to all parties involved in the food trade process. A more sustainable and dependable food trade system will be possible for everyone if blockchain technology is used to increase the transparency, security and efficiency of the food trade.

The following are some of the major advantages of blockchain technology-based certification traceability in the blockchain:Risk mitigation. By offering a safe and impenetrable method to track and confirm the authenticity of food products as they move through the supply chain, blockchain technology can help reduce the risk associated with file signing and document traceability in the food system. A blockchain-based system can verify that documents and files related to a food product that have not been tampered with by using digital signatures. Additionally, by recording all transactions and changes to a product’s status on the blockchain, it is possible to trace the product’s path from farm to consumer. Additionally, the blockchain can offer a secure, decentralized method for various actors in the food supply chain to share and access information about a product. This can help to increase transparency and trust between supply chain participants.Dramatically reduces transaction costs and time. Instead of relying on a centralized authority or middleman, a blockchain-based system enables a decentralized network of participants to access and share information about a food product. This can lower the price of using intermediaries and lessen the need for coordination and communication among various supply chain participants.Improved asset authenticity verification. Blockchain technology can offer a transparent and unchangeable record of a food product’s entire supply chain journey. This may lessen the requirement for costly and time-consuming inspections and audits. Blockchains are an amalgamation of four security features: hash functions, public–private key encryption, consensus algorithms and smart contracts.Decreased fraud and compliance costs. A blockchain-based system can use digital signatures to guarantee that documents and files related to a food product have not been tampered with, which can help identify and stop fraudulent activities. A tamper-proof record of all transactions and modifications to a food product’s status is produced by blockchain technology. This can aid in the detection and prevention of dishonest practices like the falsification of documents or the mislabeling of goods.New financing opportunities. Through the use of various techniques like smart contracts, tokenization, supply chain finance, trade finance, crowdfunding and automation, blockchain technology can provide new financing opportunities in the contemporary supply chain. By automating the payment process, these methods can increase supply chain efficiency and financing opportunities while reducing intermediaries and financing costs. They can also improve access to capital for SMEs and provide new ways to raise capital and create liquidity.Decreased risk and increased document trade facilities. Systems for tracking food products as they move through the supply chain that are based on blockchain technology may offer a more precise and effective method. As a result, product recall costs can be decreased and problem detection and resolution times can be sped up. Blockchain-based systems can offer an unambiguous and auditable record of every transaction, which can help to ensure compliance with legal requirements and cut down on the expense of compliance audits.Simple, secure data sharing between institutions. The supply chain can be made more efficient overall if blockchain technology is used to enable real-time access to the same information by multiple parties. This can help with coordination and cut down on errors. A key component of blockchain technology is consensus algorithms. They are used to make sure that new transactions are valid and that everyone in a blockchain network agrees on the ledger’s current state.

This study has shown a proof of concept of a blockchain architecture and smart contract template that can be used for international food trade certificate tracking and signing called the “GM-Ledger”. As shown in Figure 2, this solution saves a lot of time, reduces clutter and converts paper-based actions into secure digital processes.

### Role of Certifications in the Food System

Many of the manual procedures involved in the global food trade, like tracking and documentation, can be automated by using digital platforms and technologies [5]. International public and private players could have access to a wide variety of food product certifications, which helps to ensure food security for populations around the world; blockchain integration into international food trade is an essential sector for world economies. However, the conventional method of conducting international food trade is frequently slow, laborious and prone to mistakes, which can cause delays and increased costs. The need for the digitization of the global food trade is expanding in order to address these problems. The increased efficiency and speed of the process are some of the main advantages of digitizing international food trade. The time and expenses involved in the global food trade may be reduced as a result, making it more convenient and affordable for all parties involved.

The increased transparency and traceability of food products are other significant advantages of digitizing international food trade. A tamper-proof and immutable record of food products’ movement through the supply chain can be made using blockchain technology. As well as assisting in locating any potential issues or problems, this can help build trust and confidence in the food products being traded. One of the main technologies being used to digitize the global food trade is blockchain technology. A tamper-proof and immutable record can be created using the decentralized and distributed ledger technology known as the blockchain, as shown in Figure 2. This makes it ideal for use in supply chain management since it enables the creation of a transparent and auditable record of food products as they move through the supply chain.

## 2. Literature Review

A glance at both the scientific and general literature available on blockchain-based digital record-keeping will provide weighty insights into how BCT will revolutionize different industries.

“The blockchain innovation really allows us to take everything where there’s record keeping, everything where there’s trust around record keeping, and it allows us to make that digital, immutable, permanent, and global.” [6]- Jeremy Allaire

The notion of storing data in a read-only format in a distributed chain of blocks (nodes) has been around for quite some time. However, it was not until Nakamoto [7] that an actual application for recording decentralized financial transactions was made possible. Blockchains were initially seen as an alternative to financial transactions and were quickly associated with protocols such as Bitcoins, Ethereum [8] and Ripple [9]. Generally speaking, a blockchain is an electronic database that stores data digitally and is mainly used to securely record transactions using established protocols.

### 2.1. Types of Blockchain Networks

Depending on how data are stored and distributed, and who is given access to the network, blockchain networks can be classified generally into permissionless and permissioned blockchains.

As shown in Figure 3, one can further classify distributed ledger networks into the following:

#### 2.1.1. Permissionless Blockchains

A permissionless blockchain is a decentralized ledger that is open to the public. The vast majority of cryptocurrencies, including Bitcoin, are powered by permissionless blockchain networks. Here are the key characteristics of permissionless blockchains.

#### 2.1.2. Public Blockchains

The most common type of permissionless blockchain is a public blockchain. The read and write privileges are entirely unconstrained on a public blockchain [10]. The same rules that govern permissioned blockchains apply to writers. The network’s users are all anonymous. Nonetheless, some form of identity management is required when authors are given the option to remain anonymous. If not, it would be conceivable for a tiny business to impersonate a large one, giving it the ability to contribute blocks more frequently than others and, as a result, significant control over which chain of transactions is recognized as legitimate [11]. “Sybil attacks” are this kind of assault. In order to write on the ledger, writers are typically required to first demonstrate that they have completed a computationally challenging assignment. The majority of the most important cryptocurrency blockchains, including those for Bitcoin, Ethereum and Litecoin, employ this technique.

#### 2.1.3. Permissioned Blockchains

A blockchain that has been given “permission” only allows a specific entity or group of entities to write to it [12]. These organizations are the only ones allowed to propagate and validate transactions, and they also control the blockchain’s rules. The read privilege may be made available to the general public or could be partially kept secret. Here again, expensive identity management is not required because the permitted writers alternately add blocks to the chain in accordance with a predetermined algorithm. As in a private blockchain, writers on a permissioned blockchain are chastised by readers, but they are also disciplined by other writers.

#### 2.1.4. Private Blockchains

On a private blockchain, the contents of the ledger are entirely under the control of one central authority. In other words, there is just one author. The general public, the entity’s clients or a regulator might be the readers in this case. Also, different groups may have various read access on the ledger. A private blockchain does not require identity management because only one entity is allowed to write to the ledger. The system operates similarly to a privately maintained database that grants outsiders read privileges because there are no computational costs. Private blockchains are ideal for sandbox environments; however, they are not viable for enterprise or business solutions.

#### 2.1.5. Consortium Blockchains

A consortium blockchain’s main goal is to increase cooperation in order to tackle an industry’s ongoing difficulties. Consortia blockchains can be used by groups with shared objectives to restructure workflow, transparency and responsibility [13]. The consortium blockchain has a known and verified number of participants. They undertake authentication, which lowers the possibility of data threats. Nodes that go against the established protocols are promptly detected and punished. The consortium blockchain makes additional vulnerabilities like SQL injection, DDoS and “man in the middle” unimportant. The governance model states that a contract is frequently made by a relatively small number of nodes. Because it is less difficult to reach, this kind of consensus is more common. These factors have an immediate impact on transactional outputs, resulting in quick processes and enhanced scalability. Thus, mutual consensus is easier to obtain and, therefore, the blockchain does not consume much power.

#### 2.1.6. Hybrid Blockchains

A hybrid blockchain is a special kind of blockchain technology that combines elements of both private and public blockchains or aims to use the best features of both types of blockchains. As highlighted by [14], with a hybrid blockchain, businesses may create both a private, permission-based system and a public, permission-less system, giving them control over which data will be made public and who has access to them. A hybrid blockchain generates more scalability than a public blockchain network and allows for quick and inexpensive transactions.

Table 1 provides some of the references in the literature to sample the different types of blockchains and networks used to create business solutions in the food supply chain.

The following subsection briefly discusses the major features and key distinguishing features of blockchain technology. It will also discuss how blockchains will aid in streamlining the food supply chains in light of these unique properties.

### 2.2. Major Features of Blockchain Technology

Blockchain technology has four highlight features that set it apart from other ledger systems (centralized). These are provenance, finality, immutability and algorithmic consensus. Provenance refers to a full record of every transaction involving the assets that were made and stored on the blockchain. Finality, on the other hand, this means that once a transaction is committed to the blockchain, it is considered “final” and can no longer be “rolled back” or undone. Thirdly, a transaction cannot be altered, deleted or have transactions added before it after it has been recorded on the blockchain. This property is referred to as the immutability of blockchains. This feature allows the user to audit records without fear of human errors. Lastly, consensus refers to the procedure of selecting new transactions, distributing them to network users and creating a common agreement on the history of transactions.

### 2.3. Blockchain and Food Supply Chains

The food supply chain is a complex system with many nuances and intricate processes. On the whole, the food system is one of the most technologically redundant systems in the world. The major difficulties in an agri-food supply chain include a lack of mechanization, inadequate management, inaccurate information and ineffective supply chains. There is a wide body of research that suggests blockchains ought to be integrated with the food supply chains to make them more transparent, traceable and trustworthy [20,21].

Ref. [22] dives into the different pillars of food security and how blockchains can play a valuable role in the technology infrastructure of food security in a holistic sense. They also discussed the organizational, economic and management aspects of blockchain technology adoption.

As of April 2019, more than 80 brands were participating in the IBM Food Trust network, including Walmart, Kroger, Driscoll’s, Nestlé and others [23]. Other examples include the food-tracing system introduced by Chinese retail giant Alibaba in April 2018 to provide end-to-end supply chain traceability for imported goods. This consortium, called the Food Trust Framework, includes Fonterra, New Zealand Post, Blackmores and Australia Post and aims to fight food fraud and win consumer trust in the process [24]. Some of the major decentralized protocols that have use cases in the food sector are the Ethereum, IOTA and Ambrosus networks. IOTA (2016) [25] is a distributed ledger technology (DLT) that uses a unique consensus mechanism called the Tangle. The Tangle is a directed acyclic graph (DAG) that allows for transactions to be confirmed without the need for miners or fees. This makes IOTA ideal for use in applications where high throughput and low cost are essential, such as the food supply chain. Provenance, Connecting Food, FreshFarm, InFoodChain and Farm2Kitchen are some of the companies that use IOTA in their food supply chains. Bext350 uses a permission blockchain protocol called Stellar [26] that can aid in higher transactions per second. The Ambrosus protocol (2017) [27] works by creating a shared ledger of food provenance data using IoT sensor (hardware-in-place technique) data sharing. These data include information such as the origin of the food, the date and time of production, the location of each step in the supply chain and the temperature and humidity conditions at each step.

One of the many innovations introduced or amalgamated in blockchains is the smart contract. Smart contracts are codes that are executed when a condition is triggered (such as ownership change, location change, timing or crossing a value threshold). Although Bitcoin is a smart contract by definition, the idea of smart contracts for various other applications was popularized by the Ethereum Blockchain framework. These “if-else” conditions are triggered by data or events that are stored in “oracles”, which are non-blockchain sources of digital information, converting outside occurrences into information that can be accessed by smart contracts [28].

### 2.4. Tokenization of Assets

It is vital to discuss the future direction of food supply chains and their corresponding activities, namely asset tokenization. Long before blockchains, digital tokens were increasingly being used to protect sensitive information, such as personal identifiable information (PII), email addresses and account numbers. This is because tokenization significantly reduces the risk of data breaches [29]. When sensitive data are tokenized, they are replaced with a unique identifier or token. This token has no intrinsic value and cannot be used to access the original data. Tokenization has a number of benefits: It can help to protect sensitive data from unauthorized access, use or disclosure. It can make it more difficult for cybercriminals to steal or misuse sensitive data. It can help to comply with data protection regulations. The agri-food business is one area where tokenization has the potential to be used to great effect. Tokens can be used to track the movement of food products throughout the supply chain, ensuring that they are safe and traceable. Tokens can also be used to create more personalized and customized food products, based on the individual needs of consumers. In the healthcare industry, tokens are being used to protect patient records. In the financial industry, tokens are being used to protect credit card numbers and other financial information. In the retail industry, tokens are being used to protect customer data. Assets are classified based on their uniqueness (homogeneous and heterogeneous) and their nature (real or virtual) into four categories shown in Table 2. There are two kinds of assets that can be tokenized using a tokenization platform: fungible and non-fungible assets. The difference between these two assets has been discussed in the table below.

In the government sector, tokens are being used to protect classified information. Tokenization is a promising technology that has the potential to significantly improve the security of sensitive information. As the use of tokens continues to grow, it is expected to see even more innovative ways to use them to protect data. Tokenization is used to safeguard sensitive data while still allowing them to be used for commercial purposes. This differs from encryption, which involves modifying and storing sensitive data in ways that prevent them from being used for business objectives [30]. A properly constructed and executed cloud tokenization platform can avoid the exposure of sensitive data, preventing attackers from obtaining any type of usable information whether financial or personal. The important word here is “useful information”. Tokenization is not a security mechanism that prevents hackers from breaking into your networks and systems. Instead, it represents a data-centric security approach based on “zero trust” concepts [31].

## 3. Methodology

The proposed blockchain signing, transferring, auditing and storage of food certificates should be capable of capturing multiple data points about the food; high-definition photographs and financial records are both permanently recorded in the blockchain. The ownership and storage records are updated as the food changes hands, and this digital proof travels with the food as it passes through various supply chain partners. Smart contracts can be triggered by authorized retailers, warehouses, auction houses and other selling platforms to confirm food identity at any moment, protecting the food’s worth over its life-cycle. Food exporters often face concerns regarding authenticity, provenance, cultivation practices and other sanitary conditions of the food that they are importing. The data collected in the certificate signing blockchain application (dApp) have four key elements: the provenance, including its history of ownership; certificates; financial documents; trade documents; and physical characteristics such as size, appearance and weight, mode of transportation and ingredients, such as traceable content and labels.

As shown in Figure 4, the methodology this research adopts to develop a food certificate signing and archiving blockchain architecture begins by understanding the troubles and needs of key players in theory and practice. Furthermore, after designing a blockchain framework, our research will execute some key aspects of the dApp using smart contracts, which are tested using Truffle Framework in Ethereum Virtual Machine (EVM). Based on the insights from this prototype, subsequent endeavors will aim to develop an adoption framework for incorporating blockchains within the food import/export conclave of Qatar. Lastly, this study ends by pointing out future potential for maturing of this technology.

Blockchain technology has the potential to simplify certification procedures while also enhancing the supply-chain traceability of food items. Keeping track of the movement of food products from farm to fork can be challenging with traditional supply chain management systems, making it challenging to quickly pinpoint the origin of any potential food safety problems. Real-time food product tracking and easier root-cause analysis are made possible by blockchain, which enables secure, transparent recording of every step in the supply chain. In addition to lowering the risk of food-borne illness, this can help to increase the general safety and quality of food products. In conclusion, the management of trade certifications in global food supply chains has the potential to be completely transformed by blockchain technology. Blockchain can speed up the certification process and boost the efficiency and effectiveness of food safety and quality assurance programs by offering a secure and open platform for tracking and verifying information. Additionally, blockchain can enhance the supply chain traceability of food products and contribute to their overall safety and quality.

### Survey of the Similar Literature

It is vital to differentiate this research from the previously published literature. Hence, all the prior literature that meaningfully deals with certificate management and authentication is discussed in the context of their particular industries in Table 3.

Healthcare industry is one of the early adopters of blockchain technology (BCT) to help manage patient identity [32]. Notably, [33] provides a conceptual framework utilizing blockchain and smart contracts to monitor the overall oil supply chain. The work of [34,35] should be mentioned as they analyzed the area of risks and rewards of collaboration between prominent players of pharmaceutical and healthcare management.

**Table 3 foods-12-03914-t003:** Previous adoption of BCT-based certificate management in various industries.

Paper(s)	Content Analysis	Similar Publications
SmartOil [33]	Oil and Gas Data Processing	Review [36]
CertificateChain [34]	Healthcare Certificate Management	Medical Certificate [35], BHEEM [37]
Landchain [38]	Land Certificate Management	[39,40,41]
B-Cert [42]	Academic Certificate Distribution	Education Certificates [43]
TuDocChain [44]	Securing Digital Academic Certificates	Academic Certificates [45]
Halal Food Certification [46]	Authenticating Halal Food Certificate	Fish Industry [47], Palm Oil [48], Olive Oil [49]
Digital Medical Passport [50]	COVID-19 Vaccine and Immunity Certificates, Health Passports	Vaccine Certificates [51,52], NovidChain [53]
Gemiverse [54]	Certificate Management in Travel Industry	Tourism [55]
Decauth [56]	Decentralized Authentication Scheme for IoT Devices	Bubbles of Trust [57], A2 Chain [58]
Internet of Forensic (IoF) [59]	Digital Forensics	[60,61], B-coc [62], ForensiBlock [63]
CoC [64], CustodyBlock [65]	Chain of Custody of Goods Along Various Supply Chains	[66], B-Dec [67], TrueCert [68]
VANGUARD [69]	Managing Media Rights and Copyrights	[70,71], Music [72], MF-ledger [73]
**GM-Ledger** (Our Study)	**Cross-Border Food Certificate Authentication for Global Maritime Trade**	**First study to employ BCT in international food trade**

On the other hand, Refs. [50,52] analyzed the adoption of BCT to develop certificate management architecture for creating digital health passports and vaccine certificates. Blockchain was used to create more rewarding and transparent loyalty programs for travelers, according to [55]. Travelers can earn and redeem rewards more easily, and businesses can better track and manage their loyalty programs. Blockchain can be used to create more efficient and affordable insurance products [74]. One can purchase and manage their insurance policies more easily, and insurance companies can better assess and manage risk. Notably, Refs. [69,73] and many others investigated the use of blockchain technology in the field of multimedia. Their work sheds light on the importance of preventing piracy and fostering better rights and royalties in the entertainment industry. Others like [59,63] focused on utilizing blockchain technology to enhance the field of digital forensics. The primary role of such studies is to promote the use of BCT to uncover evidence related to criminal activities, such as fraud, money laundering and document tampering.

With regard to government adoption, BCT has been used by the United Arab Emirates (UAE) to streamline most of the bureaucratic processes [75]. The UAE government has launched a number of blockchain-based government services, such as the Dubai Blockchain Platform [76] and the Abu Dhabi Blockchain Strategy [77]. These services allow citizens and businesses to access government services more efficiently and securely. What is clear from this discussion is that there has not been a single part of the literature that deals with international transferring, signing and authorization of food certificates in the context of food trade. In order to shed more light on food import mechanism of the State of Qatar, the current state of food imports and the processes thereof will be presented in the next section.

## 4. Present System of Document Processing

For the purpose of narrowing down our field of study, it is good to view the entire food import process in three granular stages. These stages are arriving at the port, post-arrival and distribution. Our study is specifically concentrating on the second stage. Various certificates of the food trade and customs inspections are processed or audited after arriving at the seaport/airport. The second stage primarily involves physical inspections and certificate clearances conducted by a team of professionals with representatives from different governmental bodies, including the Ministry of Municipality (MoME), Ministry of Public Health (MoPH) and Ministry of Transport and Communication (MoTC).

With the help of field investigations and published resources, this study has defined the various steps and procedures involved in document processing and customs inspections that occur at the port of entry for food.

### 4.1. Requirement of Various Players in the Food System

This section is entirely dedicated to understanding the requirements of the main players within the food trade. Understanding their requirements and roles will help with better describing a formula for certification trading, signing and authenticating between them.

#### 4.1.1. The Requirement of Food Importers and Exporters

Corporations fuel what is known as global trade by supplying and demanding the goods that exchange hands in their roles as importers and exporters. Corporations are essential to every aspect of global trade activities from trade finance to supply chain track and trace to provenance and product pedigree. Nowadays, businesses rely on manual, paper-based, highly latent processes for international trade. The movement of goods is frequently difficult for them to see. According to the WTO [78], access to trade finance is becoming more competitive as regulation forces financiers to reduce margins and take on fewer trade finance liabilities. They also have limited capacity to track assets or identify and address emerging issues of product quality, logistics or transportation. Small- and medium-sized exporters struggle to acquire enough access to trade finance as a result of the intense competition. Importers must pay expensive insurance premiums because financiers perceive a high risk of fraud or non-delivery of goods when there is little visibility into the movement of the goods. Transparency across transactions is supported by the blockchain. This attribute provides visibility into the supply chain for use cases involving international trade. Smaller businesses will be better positioned to access trade finance products as a result of increased visibility, which will also help banks and financiers better assess the risk involved in providing financing to them. As shown by [79], businesses can use Internet of Things (IoT) technologies to gather data on their goods during shipping and store them on the blockchain.

#### 4.1.2. The Requirement of Freight Forwarders

Freight forwarders and carriers play a key role in global trade by managing the logistics of transferring cargo in supply chains. While forwarders organize the movement of the products between their origin and destination, carriers actually make the voyage. Freight forwarders and carriers engage with the bulk of their partners manually (as opposed to automatically) and employ paper-based procedures, just like other trade participants. Due to a lack of digital technology, freight forwarders frequently have to pay inflated prices, such as premium freight. The low margins and high insurance premiums faced by freight forwarders are a result of their limited visibility into the shifting of assets. Freight transporters want to prevent cargo theft. In some countries, unauthorized individuals could evade detection and collect cargo under the current system, which relies on manual verification. Due to blockchain technology, trade documents like bills of lading and purchase orders may be handled digitally. As was witnessed in pharmaceutical logistics [80], freight forwarders will be able to see the supplies they do not have access to thanks to the blockchain’s near real-time sharing and trade document verification capabilities, which will also streamline the asset-tracking process. Digital trade document management will hasten payments and interactions between trade players while reducing processing periods. One might combine document digitization and digital identification to reduce the danger of freight theft. It will be safer to use digital identities to authenticate claimant freight collectors rather than traditional ones to use blockchain to manage data collected by sensors. IoT sensors in vehicles such as trucks and other cargo ships, for instance, can be used to monitor capacity and rapidly and precisely determine the cost of volume-based freight shipments. IoT sensors are able to keep track of a product’s quality with respect to its features and requirements. This information can be shared with those in charge of quality control and recorded on the blockchain. Before implementing blockchains, carriers and freight forwarders must decide on standards for the papers they wish to digitize. They must be able to attest to the veracity of the papers they are providing when crossing borders or delivering cargo between different legal systems. The creation of standards typically involves lengthy thought and discussion with the numerous stakeholders in international trade in order for all parties to respect the digitalized documents when delivered. Freight carriers will also need to consider how and with whom they want to share their data, as well as what sort of permissions they require to match with their blockchain solutions, in order to ensure the integrity of their data.

#### 4.1.3. Port Customs Requirements

Customs and port authorities supervise and approve the transfer of commodities at junctions of transport routes. This transfer demands that layers of logistical and legal data about the cargo’s origin, destination and transit paths be sorted through. The freight supply chain has many different stakeholders, each with their own set of tools and procedures. Several versions of the same document might be confusing for end users, especially if they need to submit the same information in various places. Such circumstances expose the supply chain to dishonest or malicious behavior. Many processes, including customs clearance, still call for paper documentation and manual completion. Intermediaries often spend hours just keeping track of the paperwork required to finalize deals. These processes are inherently vulnerable to human error. If trade documents were digitized and stored on a distributed ledger, customs and port authorities would have quick access to and knowledge of regulatory standards, clearance status, import and export certificates, classification information, origin information and tariff codes. As witnessed in Dubai government implementations [81], digitized trade documents on the blockchain would improve trade facilitation and customs enforcement, reducing operational costs and the amount of human paperwork needed. Asset tracking capabilities on the blockchain would also provide customs and port authorities with greater visibility and insight into the current and anticipated flow of products, helping them to better plan for demand and improve operating efficiencies. The majority of port authorities presently engage with their supply chain partners, including companies and governmental entities, using electronic channels known as port community systems (PCS). PCS offer transport network management services. For transportation and trade regulations, respectively, PCS connect to marine single window systems and trade single window systems in the most cutting-edge trade communities. For ports and customs authorities to fully utilize blockchain, any platform built on it must be able to interact with these systems. These processes are inevitably vulnerable to errors caused by humans. If trade documents were digitized and stored on a distributed ledger, customs and port authorities would have quick access to and knowledge of regulatory standards, clearance status, import and export certificates, classification information, origin information and tariff codes. Digitized trade documents on the blockchain would improve trade facilitation and customs enforcement, reducing operational costs and the amount of human paperwork needed.

#### 4.1.4. The Role of National Regulators

As witnessed in the Dubai Government [76], blockchains are employed to transform governments to expedite and track digital paperwork without suffering a waste of time. In Qatar, multiple national organizations are involved in securing the food system and ensuring its safety and quality. The Ministry of Municipality and Environment plays a key role in regulating food imports, focusing on health certificates, bills of lading, invoices, and packing lists to inspect imported food products. The Ministry of Public Health (MoPH) also participates in regulating food imports, establishing safety standards, inspecting imported products, assessing the market supply, and collaborating with various government bodies to enforce relevant laws and ensure food security. These two organizations work closely together to uphold food safety and quality standards.

Additionally, the Ministry of Transportation and Communications (MoTC) manages the logistics and transportation of imported food products, overseeing ports and airports, coordinating cargo movements, and collaborating with other government entities like Ministry of Municipality and Environment and the Ministry of Public Health to ensure compliance with all relevant laws. The Ministry of Transportation and Communication also focuses on the safety and quality of food products during transportation, coordinates transportation infrastructure, and collaborates with relevant organizations to ensure efficient and secure food transportation. Overall, Qatar’s food security is a result of the concerted efforts of these national organizations to enforce strict standards and regulations for food imports. In the following subsection, the role of insurance providers is discussed.

#### 4.1.5. The Role of Insurance Providers

The protection of food imports and exports that are shipped by sea is significantly aided by insurance companies. Providing coverage for the various risks connected to shipping food products by sea is the primary responsibility of insurance companies in this situation. The following are a few of the risks that insurance companies may cover:Insurance companies can cover damage or loss to cargo that occurs during transportation, such as that caused by inclement weather or ship mechanical issues.Insurance providers can offer coverage for any third-party claims that may result from the import or export of food products, such as if a product is discovered to be contaminated or if a consumer experiences an allergic reaction.Insurance companies can cover any delays or interruptions that may happen during transportation, such as when a ship is held in port because of a quarantine or when it is delayed by bad weather.Insurance companies can offer coverage for any damage or loss that may result from war or strikes, which can obstruct the movement of goods.Additionally, insurance companies may offer coverage for the food items both during transit to and from the destination, as well as during warehousing and storage there.

As proven by [74], blockchain technology has a vital role to play in the field of insurance coverage. It is crucial for importers and exporters of food products to work with insurance companies that have experience in the food sector and can offer protection against the specific risks involved in shipping food products by sea. Given the entire cycle of cross-border trade, both importers and exporters must safeguard themselves against currency fluctuations. Despite strong justifications for insurance, exporters and importers throughout the supply chain do not directly obtain insurance. They work with specialized brokers to insure their products, but the majority of insurance in international trade happens between insurers and banks (for applications involving trade financing) or between insurers and freight carriers (for applications involving track-and-trace), and it spreads along the supply chain as a result. Insurance companies must go through a protracted claims management process before they can pay claims. During this process, they must reconcile different versions of the same information that were recorded in documents that were created independently by clients, brokers, retro- and re-insurers, underwriters and claims managers. It takes a lot of time and money to match documents. The resolution of disputes between insurers and claimants is frequently complicated by this lack of certainty. Payment on a claim can occasionally take many years to process. When goods are damaged or destroyed somewhere in the chain of custody, it can be challenging to determine which insurer is responsible in some applications where different supply chain participants are clients of different insurers. According to the information at their disposal, insurers also typically absorb additional costs by underwriting more frequently for maximum risk rather than actual risk. Blockchain-based processes are more tamper-proof than paper-based procedures.

### 4.2. Document Processing Model

Based on the above study of the roles and requirements of the various players in the food system, this study has created a business process model diagram to show the orderly movement of information and documents. Figure 5 below shows the business process diagram involved at every stage of vessel docking. This process involves the following:

The importer identifies potential food products to import and reaches out to the exporter to negotiate terms of trade.The exporter provides the necessary documentation and certifications required by the importing country’s regulations.The importer arranges for a freight forwarder to handle the logistics of exporting the products, including booking shipping and customs clearance.Exporters ship the products to the importer’s designated port of entry.The freight forwarder handles customs clearance and arranges for transport to the importer’s facility.Regulators, including government ministries and International Food Standards documents, have to be verified.Importers conduct necessary inspections and obtain any required permits or licenses from regulatory agencies.The importer arranges for insurance coverage for the products.The importer distributes the products to retailers or wholesalers.

This process model will aid in coordinating the infrastructure and logistics of the maritime food trade systems to make sure they can handle the volume of imports of food. This system is guaranteed the efficient and secure transportation of food products.

## 5. Proposed Document Processing Procedure Using Blockchain Technology

The global food trade network is plagued by four issues. The confidentiality of transactions comes first. The integrity of data is the second. Repudiation of facts comes in third place, and authentication comes in fourth. All of these four issues are solved by applications of cryptography. This concept is summarized below based on the work conducted by [82].

Confidentiality is achieved with the aid of cryptography. In this system, a receiver’s (example: the importing country’s customs official) public key is used to encrypt the message (metadata), and then, as soon as this message is received at the receiver’s end, the receiver will decrypt the message with the aid of their own private key. This method is secure and time-tested. Nobody else can decrypt the message because no other entity has the receiver’s private key. Similarly, the authenticity of certificates is achieved using a cryptographic technology called digital signatures. A digital signature is based on cryptographic technology, which provides greater document security. An electronic signature offers no document security because it lacks a document verification process and no way to track changes made to the document’s content after signing. Blockchains can be used to transfer, transact, verify and sign certificates in a secure, transparent, efficient and cost-effective way using digital signatures [83]. This is because blockchains are a tamper-proof distributed ledger technology that can store and track data in a secure and transparent manner. The data on the blockchain are encrypted and distributed across a network of nodes, making it difficult to hack or tamper with. Additionally, anyone can view the transaction records on the blockchain, which makes it easy to verify the authenticity of a certificate.

As shown in Figure 6, the system of “GM-Ledger” certificate authentication can authenticate and track food certificates without compromising security and speed. The officer at the customs port who wants to validate and authorize the import of food materials and containers can simply verify their signatures and trace their origins to respected government authorities using a smart contract that can access the files that are stored in a blockchain-hosted Interplanetary File System (IPFS). Further below, the structure and design of the smart contract used for this purpose are discussed.

### Proposed Blockchain Solution Architecture

In the Ethereum Network, a block is a bundled piece of data that contains both information about the consensus process and a chronological list of exchanges. Blocks are put forth by Proof of Stake (PoS) validators, after which they are distributed throughout the entire peer-to-peer network, where all the other nodes can quickly and independently verify them [84]. Any invalid blocks are ignored by the network because consensus rules define what constitutes a valid block and what is not. This network’s current state is represented at the end of a deterministic chain of events that is created by the ordering of these blocks and the transactions contained within them. Next, in order to design a blockchain architecture, the practical implementations in past research publications are examined in Table 3. This work is proposing our blockchain architecture to be based on the Ethereum Network. In blockchain technology, the term “layers” refers to the different levels or components that make up a blockchain system [85]. Each layer performs a specific function and serves a specific purpose and, together, they form the overall structure of the blockchain. Each layer that makes up the Ethereum Blockchain has a distinct function and goal. These layers consist of the following:The top layer of the Ethereum Blockchain is called the application layer, and it is here that users interact with decentralized applications (dApps) and smart contracts. User input management and the presentation of smart contract execution results are handled by the application layer.The contract layer is in charge of carrying out smart contracts. Smart contracts are self-executing agreements in which the terms are directly written into the program code. A contract can be automatically executed by the network once it has been deployed on the Ethereum Blockchain.The message layer is in charge of managing the dialogue between the application and contract layers. The input is transferred from the application layer to the contract layer, and the output is transferred back to the application layer for display.The integrity and security of the network are upheld by the consensus (or blockchain) layer, which is the foundational layer of the Ethereum Blockchain. It is in charge of maintaining the distributed ledger and using consensus algorithms to confirm and validate transactions. It is responsible for verifying and validating transactions and for maintaining the distributed ledger.The P2P/data network layer is in charge of the upkeep of the network infrastructure supporting the Ethereum Blockchain. It is in charge of propagating new blocks to all network nodes and the upkeep of the peer-to-peer network that underpins the Ethereum Blockchain. This is the bottom layer of the blockchain and is responsible for storing and managing the data that are recorded on the blockchain.

Figure 7 summarizes all the above discussion to show the various layers of the proposed blockchain-based solution. Each layer has a unique set of technologies and protocols that work together to support the entire Ethereum Blockchain ecosystem. This modular design allows for flexibility and scalability, as different layers can be modified or replaced without affecting the entire system.

Anyone can use IPFS to be both a client and a server, with about equal effort in each case. The IPFS not only combines the best Internet services, protocols, layers and architectures into a single architecture but also enables a smooth migration by ignoring all the patched and ineffective bits that are typically kept when switching to a newer technology. The IPFS provides:A procedure for tracking down content and organizing delivery from one location to another.A file system that can be mounted locally so that remote resources can be accessed as if they were local.A modular method for thinking about network operations like virtual circuits and routing.File transfers between peers that do not require servers.A global namespace built on the Public Key Infrastructure (PKI).A method for maintaining the consistency and version control of files.An upgrade path for browsers so you can access information using the new way (ipfs://) or the old way (http://).

IPFS may be summed up as distributed, verified and hash-linked data structures. The building blocks of blockchain data structures are, likewise, hash-linked data structures. In the case of IPFS, one can store bigger files (PDFs, videos, documents, pictures) as opposed to a simple blockchain network that stores transactional data only. The following section will discuss how BCT can be adopted in the context of a wider supply chain.

## 6. Smart Contract Design

At the core of certificate transactions is the role of smart contracts. Smart contracts, which allow for the execution of tasks, like transactions, updating wallet balances and retrieving ownership data, are the engines behind decentralized applications. End users receive an interface from the dApp. Automating the certification process is another way that blockchain can be used to simplify food trade certifications. In many instances, certifications are granted based on the fulfillment of particular requirements or the submission of particular paperwork. Blockchain technology enables the automation of this process by utilizing smart contracts, which are contracts that execute themselves in which the details of the agreement between each of the parties are directly encoded into lines of code. By automatically confirming that the necessary requirements have been met, and issuing a certification once those requirements have been met, smart contracts can be used to automate the certification process. This can speed up certification processes and lower the possibility of mistakes. For insurers, the legitimacy of smart contracts will be crucial, especially during the claims management process, which calls for multiparty agreements on ancillary contract documentation.

Figure 8 discusses the design of a smart contract for signing certificates using cryptographic signatures and consensus. Using smart contracts in the global food trade is one of the main blockchain applications. Smart contracts are agreements that automatically carry out their terms after being written into computer code. This makes it possible for contracts to be automatically executed based on pre-established conditions, which can lessen the need for manual intervention and speed up the procedure. The use of digital identities is a further blockchain application in the global food trade. All parties involved in the food trade process can have their identities securely and impenetrably recorded using digital identities. This can aid in boosting trust and confidence in the food products being traded and aid in spotting any potential issues or problems.

The following solidity contract implements a signing contract. Of course, the main problem of certificate electronic signing is how to assign voting rights to the correct persons and how to prevent manipulation. The plan is to construct one contract for each signing session, with a short title for each choice, as shown in Appendix A.

The right to sign will then be granted to each address separately by the contract’s originator, who also acts as chairperson. The persons whose names are behind the addresses can then decide whether to sign themselves or to designate a signer they can trust. MostSignedCertificate() will return the certificate with the most signatures at the conclusion of the signing time and, hence, help us identify and reward trustworthy importers and their corresponding value chains. The following section will discuss the execution of the smart contract and the cautions that need to be adopted when creating a smart contract.

## 7. Results

The created solidity contract was uploaded to a testing environment called RemixIDE. For creating, deploying, troubleshooting and testing smart contracts compatible with Ethereum and the EVM, the Remix Online IDE was used. It has no setup requirements and has an adaptable, simple user interface.

The solidity contract was successfully compiled using the IDE. A static analysis of the contract was furnished. The first one was concerning gas limits. The gas requirement of the function **Signing.delegate** is infinite: if the gas requirement of a function is higher than the block gas limit, it cannot be executed. The IDE suggested that we avoid loops in the functions or actions that modify large areas of storage (this includes clearing or copying arrays in storage).

The next finding was regarding the use of loops. Caution must be exercised when using loops that do not have a set amount of iterations, such as those that rely on storage variables. Transactions are limited in how much gas they may use by the block gas limit. The block gas limit may be exceeded by the number of iterations in a loop, which might eventually cause the full contract to halt. Additionally, utilizing unbounded loops results in high unnecessary gas expenses. The information on how many objects one can pass to such functions at once to ensure success should be tested with care. Since this study is a preliminary study on blockchains and their uses in the food supply chain, we intend to rectify this issue in the next iteration of this research.

### 7.1. Advantages of Tokenization for Players

There are several advantages to tokenization in the processes and procedures that are involved in the supply chains of food and how they are implemented by the players of the food supply chains. These advantages will help build policies for player interactions and also help in the propagation of novel procedures for supply chain management:

**Enhanced liquidity:** A non-liquid asset’s liquidity is increased by blockchain-based asset tokenization [86]. Let us use the scenario where a shipping company requests QAR 500,000 be taken out of a ship with a QAR 50,000,000 valuation. This shipping firm might have tokenized its shipping vessel into 500,000 security tokens, each worth 0.0002 percent. In order to ensure a more liquid asset, they might sell just over 5000 tokens as opposed to selling the entire ship and eliminating its use as a mode of transportation.

**Fair prices:** There is frequently no established market price for assets that cannot be sold. In this situation, asset owners frequently offer incentives to buyers, such as non-liquid discounts, which lower the asset’s price. Because fractional ownership is made possible by tokenization, non-liquid discounts are eliminated, increasing an asset’s liquidity. Additionally, owners are able to charge a fair market price by selling tiny fractions of ownership [87].

**Decreased management costs:** Today, it takes more time and money to transfer ownership of an item since you need attorneys to handle the paperwork and establish confidence with the buyer [88]. Many steps of this procedure will be automated if sellers opt to tokenize the exact same item and use a decentralized platform or marketplace, which will save time and money.

### 7.2. Advantages of Asset Tokenization from the Viewpoint of Investors

The following are some of the advantages of tokenization to the investors of the food supply chain (FSC):

**Increased portfolio diversification:** Retail investors are now able to invest smaller sums of money in a shipping company, as seen in the previous example of a tokenized ship. By investing, for instance, a sum of QAR 10,000, investors have the opportunity to diversify their portfolio. Historically, without a lot of documentation, which costs money and requires extra time, this would not have been feasible. Because of asset tokenization, investors gain from increased asset liquidity.

**Shorter lock-up times:** Investors cannot sell their assets during lock-up periods. This can sometimes be attributed to the asset’s size and non-liquidity. The tokenization of assets may shorten the lock period since investors may immediately sell their tokens on a liquid market. In this case, investors do not have to wait years before taking profits or losses.

**Open process:** Owners are unable to alter an asset’s history to make it more appealing because the blockchain’s asset tokenization underlying technology is immutable. Investors can use this information to see a holding’s history and make better decisions.

**Secure identity:** As illustrated by [32], for purposes like Know Your Customer (KYC) verification, a buyer’s private–public key pair creates a digital signature proving they are who they claim to be. The blockchain stores information on ownership and decentralized identity (DID). There are also DID identifiers chosen by standards bodies, like W3C, which guarantee acceptance across numerous networks and platforms.

**Asset tokenization’s foreseeable future:** The current model of asset management is about to be completely changed by tokenization. It guarantees security and fairness while democratizing market access. Legal restrictions are currently the only roadblock, and how much of a barrier they are will depend on the kind of asset someone wants to tokenize. To resolve tax-related and cross-jurisdictional issues, building a judicial link between assets and decentralized ledger technology experts is required. In spite of this, brand-new products will eventually enter the market and resolve these issues.

### 7.3. Role of Blockchains in the Future of Supply Chains

In the future, this study proposes to see such applications of the blockchain in the high-valued goods and services industry. These include wine, caviar, diamonds, cancer drugs, vaccines, pearls, etc. It is also clear that such tracking applications require two prior conditions. One, the condition of perceived value, and two, the condition of susceptibility to adulteration. Here are some important considerations regarding the role of blockchains in the future of supply chains:Organizations’ adoption of the blockchain is still not fully understood. When they are fully integrated into an organization’s system, blockchains typically shine the brightest. Organizations’ need to address compatibility issues may increase as the number of off-chain components increases. Since this is challenging to accomplish, it is preferable to start with a specific area, like certification transparency.Both data quality and immutability are crucial. The blockchain is unique among data processing technologies due to the immutability of the data. But, data entered into supply chains are frequently inaccurate because people make mistakes. Every data-related action is treated as a transaction on a blockchain. On a blockchain, data can only be updated; it cannot be fixed traditionally. Organizations will need to process more transactions and expend more resources processing more updates, which will result in more transactions.Giving the appropriate people access to data. Granular granting of data access rights is one method of addressing the prior problem. Sharing crucial data with the appropriate parties and in the appropriate situations is crucial to preventing data leaks and monetary losses. Organizations should decide on various levels of data confidentiality to ensure that no unauthorized users, including third parties, will access information they are not supposed to. Each user of an organization’s supply chain management system can be given a specific role with corresponding access rights.Develop this architecture in the correct way. Building a blockchain-based supply chain management solution typically takes one of three common approaches:Make use of an established global platform. Because they have been extensively tested by a large number of users, large, well-known blockchain platforms are typically trustworthy. A ready-to-use platform, however, might offer few opportunities for customization and necessitate substantial changes to an organization’s current system.Make use of a public blockchain that has smart contracts. Making a smart contract-based solution that meets an organization’s needs is a fairly simple process. Public blockchains that are widely used support smart contracts, however, they may be too slow or expensive for supply chain tasks.Create a unique network. A customized blockchain network can operate flawlessly, fit a business’s existing system perfectly and have affordable transaction processing costs. However, creating such a solution requires a high level of expertise, careful planning and extra work.Several parties, including producers, suppliers, merchants and final consumers, are linked by supply chains. As a result, they rely on a variety of tools, including tracking systems, management software and Enterprise Resource Planning (ERP) systems. Using all of these tools to integrate a blockchain platform can be very difficult. In the first place, not all third-party programs and systems support blockchain technology. The architecture of an organization’s solution, including all of the Application Programming Interfaces (APIs), containers and micro-services an organization employs, has to be properly thought out. Organizations should give their full attention to the security of data, both in transit and at rest. Second, it is possible that not all of an organization’s suppliers, partners and customers would agree to utilize its platform if it employs blockchain technology. They could be concerned about anything from system integration costs to data security issues while using a shared environment. As can be seen, despite their many benefits, supply chains built on blockchain are not that straightforward to put up. Organizations may fully enjoy the advantages of blockchain-powered supply chain management by taking the aforementioned difficulties into consideration and carefully organizing their operations.Blockchains must be connected to IoT devices to guarantee smooth feedback and prevent data manipulation. But, blockchain’s lack of scalability is the issue. Particularly, it is known that, in their present forms, public blockchains, like Bitcoin and Ethereum, process between about 7 and 25 transactions per second on average. Due to the proof of work, these public blockchains have distributed consensus requirements by design, which makes them fatigued and concurrency-limited. As a result, public blockchains stamp micro-transactions with exorbitantly high levels of fees. For a USD 1–2 transaction, the median fee at the time of writing was 34% for Bitcoin and 8% for Ethereum [89].

There are some BCT solutions, such as the Ambrosus project, which uses Ethereum to instrument and sense the food chain. Instead of erratically promoting micro-transactions (such as the temperature readings of a pallet of strawberries) to the Ethereum Blockchain, Ambrosus seeks to massively aggregate these events before coordinating with the blockchain and to store only references to events, pushing the actual events to a decentralized database [27]. This method boosts the system’s feasibility for IoT sensing use cases, but it is unable to solve use cases that need regularly updated data or latency issues with actuation use cases. For this reason, Ambrosus is concentrating primarily on using IoT to instrument the supply chain: a use case that is centered around sensing.

Blockchains can be used in tracing financial fraud [90]. Corporations have set up numerous local trade finance programs with significant duplication inefficiencies and very little standardization across organizations. As a result, the global treasury has limited visibility to effectively optimize working capital compliance and counter-party risk across entire organizations. By utilizing blockchain technology, TradeX, AIG and Standard Chartered work with organizations to develop solutions for the financial and physical sides of the supply chain. This allows for expanding our ecosystem securely and effectively.

The following section will discuss the future research pathways that one can draw from blockchain technology adoption and how it can complement the existing verticals and priorities of Qatar’s food security program.

## 8. Future Works

The blockchain still faces a lot of drawbacks, such as scalability and network security [91]. When it comes to adopting certificate authentication using the blockchain network, there are other areas where future work needs to be concentrated.

### 8.1. Network Architecture

The theory of operations research can help design and optimize the network architecture for a blockchain-based supply chain management system. This can involve analyzing factors such as the number of nodes in the network, the geographic distribution of nodes and the use of sharding to improve performance. The network architecture of a blockchain-based supply chain management system refers to the way in which nodes (computers or servers) are connected and communicate with each other. The design and optimization of this network architecture can significantly impact the performance and scalability of the system.

The number of nodes in the network can impact the system’s performance in multiple ways. A higher number of nodes can improve the system’s security and decentralization, but it can also increase the processing time and reduce the scalability of the system. Operations research can help identify the optimal number of nodes required to balance security and scalability. The geographic distribution of nodes is also an important factor to consider in designing the network architecture. Having nodes in different locations can reduce latency and improve the overall performance of the system. Operations research can help identify the optimal geographic distribution of nodes based on the specific supply chain management use case.

### 8.2. Complete Blockchain-Based Tokenization of Food Supply Chains

As mentioned earlier, a token is a piece of information that functions as a stand-in for a more valuable piece of data. Tokens have almost no intrinsic value; they are only useful because they represent anything larger, such as a credit card or a bill of transaction. In the agri-food sector, tokens represent distributed ownership of the underlying asset’s value. This means that multiple parties can own a token, which can democratize the process of ownership. Blockchains have a significant role in the way individuals or nations invest, evaluate and exchange physical assets, like land, sensitive documents, food and automobiles [92]. Tokenization is a solution that transforms an asset into a digital asset. As shown in Figure 9, depending on the tangible or intangible nature of these assets, tokens can be studied as four different types. Every digital token acts as an ownership share. One can use smart contracts to tokenize their assets.

The food supply chain consists of various activities such as production, procurement, inventory management, distribution, transportation, consumption and quality control. Many of these activities are either fragmented or highly centralized, thus creating silos of knowledge within organizations and wealth hoarding by a few powerful corporations. This work shows that the responsible use of tokenization and its adoption by governments can help resolve many of these issues. Tokens are generally categorized into security, utility and payment tokens [93]. Blockchain-based tokenization refers to the digitalization of a real-world asset while making sure its ownership and value are immutably stored in a decentralized ledger. This concept has spawned a myriad of possibilities in businesses with regard to the liquidity of assets. Blockchains enable better property records in emerging markets and offer the ability to make everything a tradable asset. Tokens also bring with them the concept of brand loyalty programs, customer reward systems and value creation. Tokens have the ability to behave as multi-faced access passes as well. With the onset of blockchains, non-fungible tokens have become the new buzz word and they carry along with it a wide array of applications. For example, Gary Vee, in 2007, released non-fungible tokens that acted as ownership tokens, brand value tokens and also as tickets to his international VeeFriends Conference [94]. These tokens ensure that transparency is maintained across the system as blockchains are essentially a global ledger that stores all access transactions. It is relevant to mention the study of Nir Kshetri [95], which identifies four different categories of the applications of non-frangible tokens.

### 8.3. Roadmap for Developing Human Capacity

Firstly, national governments and corporations have to raise awareness, resources and commitments to action. All around the world, governments are building think tanks and research centers for blockchain research and policy framing. The UK and the US have national-level bodies, such as the All-Party Parliamentary Group on Blockchain (APPG Blockchain) [96] and the National Institute of Standards and Technology (NIST) [97], respectively, for regulating and studying blockchain computing and its implications to society at large. To meet the significant and cumulative technological learning requirements of blockchain governance for food systems and related supply chains, the national innovation system must be reoriented. Individuals and firms need to make sure that they resolve coordination failures, attract complementary investments and leverage network effects to use the blockchain as an empowerment and service delivery infrastructure.

Secondly, the nation has to build alliances for combined action for policy and institutional redesign. Some of the methods used to procure, trade, vote, sign and certify food and consumption need to undergo significant organizational transformation. Humans are at the center of every digital revolution; therefore, significant strides must be undertaken on this front.

Thirdly, there is a need to clarify duties, nurture participation and establish public–private partnerships involving all parties, including NGOs in the food system. A national plan should aid in the clarification of roles and functions, as well as the facilitation of wide participation in the development and execution of important initiatives. It should not be seen as solely a government plan; on the contrary, it should be a joint effort. It should describe the government’s responsibility in establishing regulatory and institutional frameworks, as well as in promoting blockchain technology to private firms and civil society. Strong mechanisms must be put in place to support market dynamics, promote social applications, enable bottom-up efforts and ensure shared learning and scaling up. Fourthly, there must be a focus on exploiting blockchain technology for national food security objectives, as well as assisting in the sequencing and phasing of complementary expenditures; policymakers and other stakeholders can use a national technology adoption plan method to target, prioritize, sequence and phase investments and complementary actions. It should encourage investment and complementing measures through partnerships. This is especially important in the case of e-government, institutional paradigm shifts, long-term commitments in public–private collaborations and other public-sector applications that need large expenditures. Likewise, it will be necessary to establish objectives for improving access to information infrastructures for organizations, people, schools, government organizations, civil society and the scientific community. Without such national strategies, data system investments are frequently donor-led and fragmented, leading to priority distortions, enclave activities, the duplication of investment opportunities and a dilution of efforts, as well as unrealized or unviable benefits and limited scaling-up opportunities.

## 9. Conclusions

In conclusion, we have discussed the possibilities of cooperation in the food system, particularly in Qatar, the necessities of certifications and standards in the food system, the import-export sector of the food system of Qatar, the prospects of blockchain technology, its benefits and also proposed a novel idea to sign, audit and track food certificates across the food system.

In summary, transaction processing is a key factor in the performance and scalability of blockchain-based supply chain management systems. Operations research can be used to optimize transaction processing by analyzing transaction sizes, transaction fees and the use of off-chain solutions. This improves performance, reduces transaction fees and increases scalability. Off-chain solutions are techniques used to offload the blockchain network by processing transactions outside of the blockchain network. Operations research can help determine the best use of off-chain solutions based on factors such as transaction volume and specific supply chain management use cases.

This study illustrates that blockchains are a better alternative to the existing traceability solutions that promote a silo mentality and inefficient collaborations. The blockchain has the ability to assist governments to minimize fraud, maximize supply chain stakeholder participation and champion paperless-digital operations while also enabling cooperation across many divisions and branches to offer residents more efficient and effective services. Furthermore, the implementation of the blockchain might enable government agencies to deliver new value-added services to businesses and others, perhaps generating new income streams for them. The blockchain revolution is going to enable access to information transparently and responsibly toward public empowerment and, thus, alter the mechanism of food system governance. We also saw that blockchain technology can be viewed from an organizational strategy fit viewpoint and knowledge management point of view. Establishing knowledge-rich settings entails not just ensuring transparency, but also ensuring that a diverse range of views and issues are heard and properly handled. With its four main characteristics—decentralization, provenance, job automation and auditability—the blockchain has demonstrated its potential to revolutionize established industries. This study provides a thorough analysis of the nationwide deployment of blockchain technology for food security.

## Figures and Tables

**Figure 1 foods-12-03914-f001:**
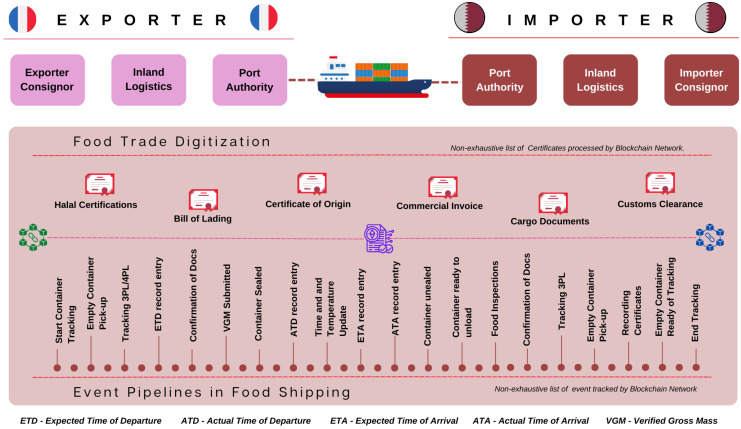
Digitization of trade certificates in the food system.

**Figure 2 foods-12-03914-f002:**
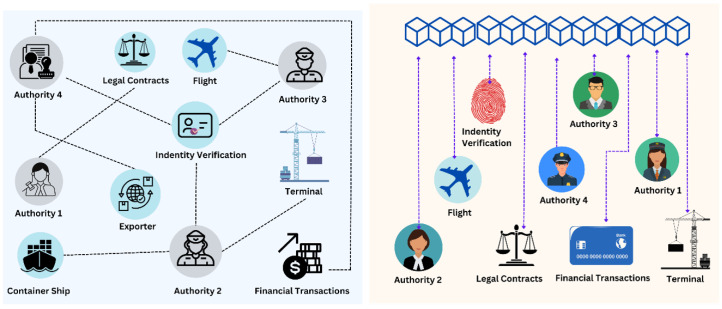
Present system versus proposed system: (**Left**) Present information sharing system without blockchain technology (**Right**) Proposed system with the adoption of blockchain technology.

**Figure 3 foods-12-03914-f003:**
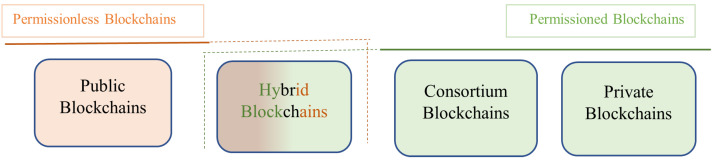
Types of blockchain networks.

**Figure 4 foods-12-03914-f004:**
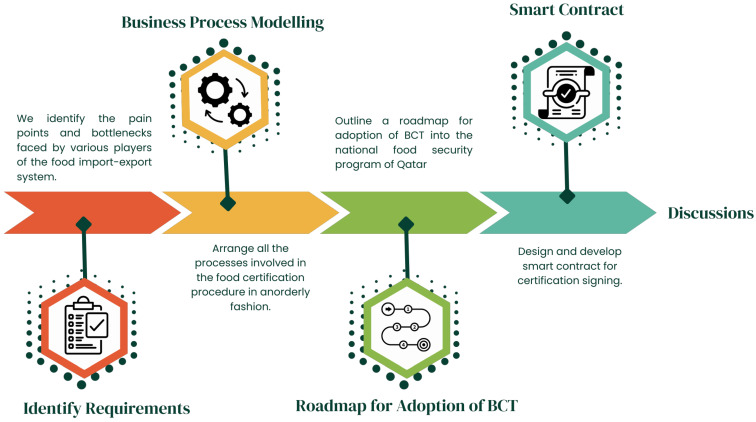
Methodology to find a BCT-based solution to certificate trading.

**Figure 5 foods-12-03914-f005:**
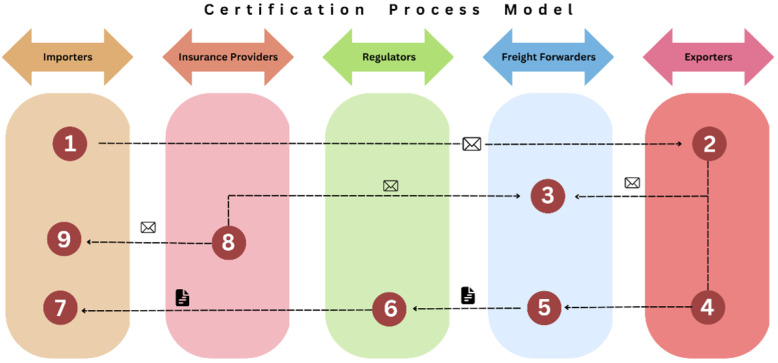
Document process diagram.

**Figure 6 foods-12-03914-f006:**
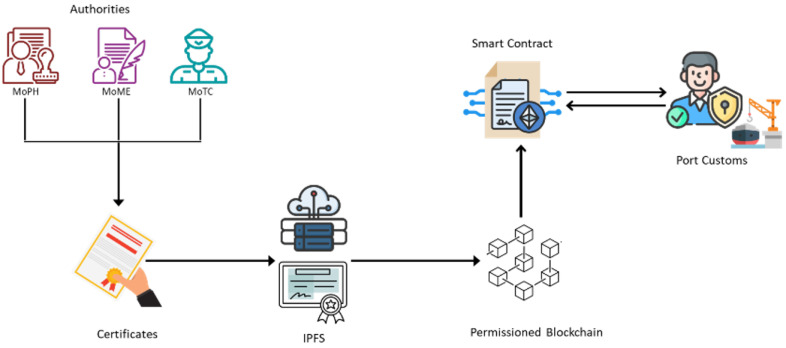
GM-Ledger certificate authentication.

**Figure 7 foods-12-03914-f007:**
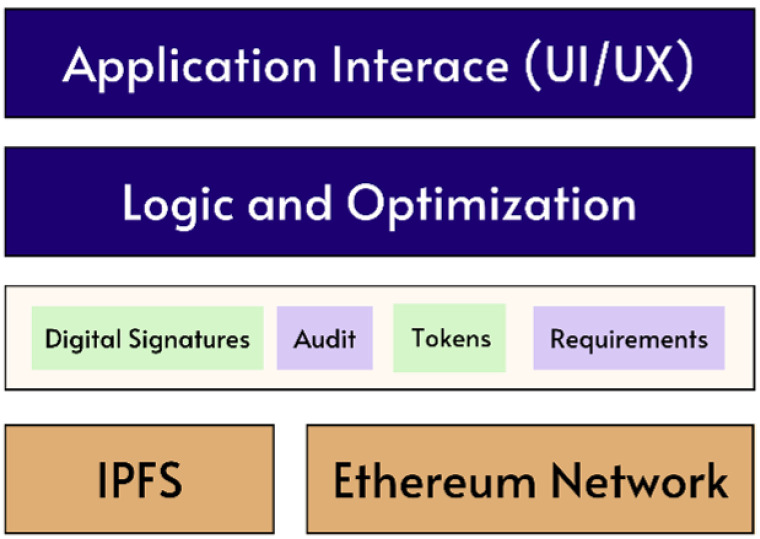
Technology layers of the proposed BCT solution.

**Figure 8 foods-12-03914-f008:**
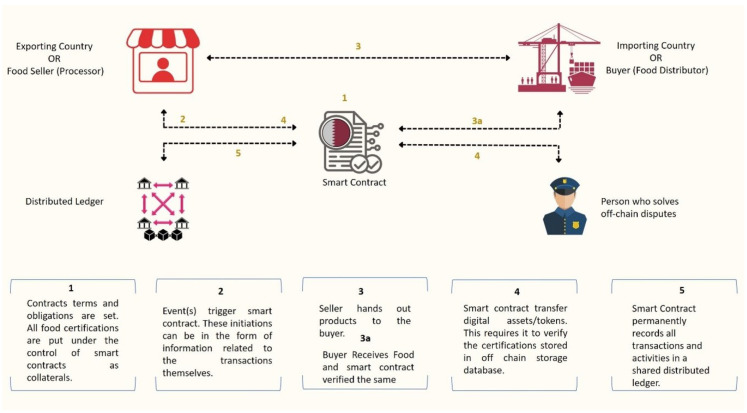
Design of smart contract for food certification management.

**Figure 9 foods-12-03914-f009:**
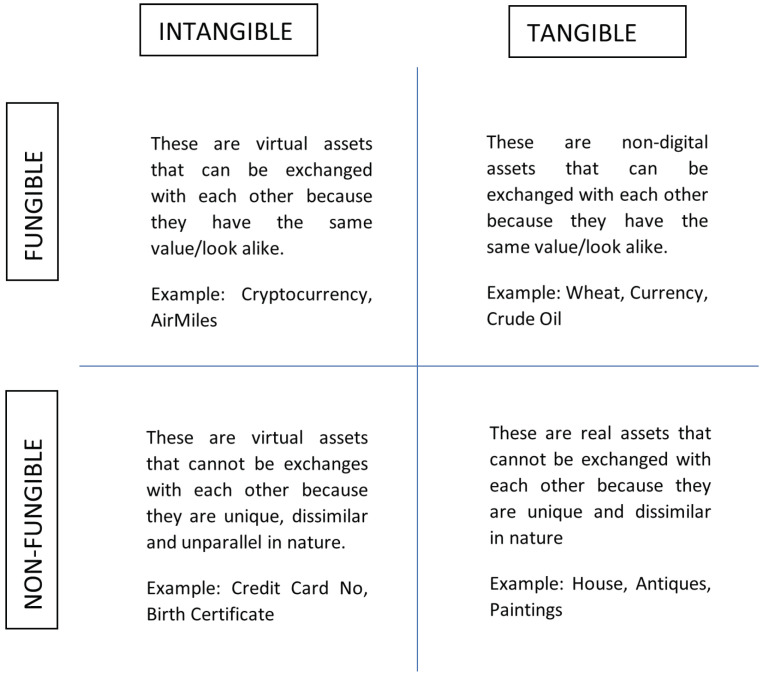
Different types of tokens.

**Table 1 foods-12-03914-t001:** Various blockchain networks used in food supply chains.

Value Created	Type of Blockchain	Blockchain Network	Reference
Traceability of food	Permissioned—private	Hyperledger Fabric	[15]
Consensus toward buyer–seller match making	Permission less—public	Ethereum Network	[16]
Developed three algorithms for better management of the FSC	Hybrid AWS	Ethereum Blockchain	[17]
Detailed information about the different blockchain networks	Permissioned—consortium	Hyperledger Sawtooth and Fabric	[18]
Conducts feasibility study of blockchain adoption for food safety	Permission less—public	Ethereum Network	[19]

**Table 2 foods-12-03914-t002:** Difference between fungible and non-fungible tokens.

	Fungible	Non-Fungible
Interchangeability	Interchangeable: A token can be exchanged to any other token of the same type.	Not Interchangeable: Non-fungible tokens cannot be replaced with another non-fungible token of the same type.
Nature	Uniform: All tokens of the same type are identical in specification, each token is identical to another.	Unique: Each token is unique and different to all the other tokens of the same type.
Divisibility	Divisible: Fungible tokens are divisible into smaller units, and it doesn’t matter which units you get as long as the value is the same.	Non-divisible: Non-fungible tokens cannot be divided. The elementary unit is one token and one token only.

## Data Availability

The datasets generated for this study are available on request to the corresponding author.

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
