# Peer review of "GM-Ledger: Blockchain-Based Certificate Authentication for International Food Trade"

_foods, 2023, doi:10.3390/foods12213914_

Round 1

Reviewer 1 Report

authentication is one of the major techniques to restrict the unwanted use doing any activity in an application. In this work, the authors proposed blockchain technology to enable a framework to address the challenges in the International Food Trade system. The overall contribution and presentation of the work are good. There are some comments which will improve the paper which is mentioned below:

Major comments:

1. The abstract is not clear, and some of the sentences are incomplete. Do a proofreading and change accordingly.

2. The authors must mention their contribution at the end of the introduction section.

3. Initial part of the literature survey authors used BCT What is that? And lines 124-127 authors must give the reference.

4. Line 199 looks ambiguous

5. Line no 271 is wrong, table 4 not Figure 4.

6. In Section 3 Many times authors mention I, for example:  I adopt,   I  will.....

rectified these incorrect sentences.

7. Section 5, Figure 8 caption is wrong which authors claim as their proposed work as " authentification".  The line 541 authors claim this is new term but in my knowledge some of the authentication using Blockchain are already their. Please look into these research paper like a) Bubbles of Trust: A decentralized blockchain-based authentication system for IoT

b) Decentralized authentication and access control protocol for blockchain-based e-health systems

c) Decauth: Decentralized authentication scheme for iot device using ethereum blockchain

8. The result section can be improved using some critical analysis  in term of efficiency/reliability/etc of the proposed system

9. The one comparative statement may be qualitative or quantitative analysis is more meaningful to end this paper.

Minor comments:

1. Do check the manuscript thoroughly to avoid typo errors.

2. As mentioned use some relevant research paper related to your problem statement which is authentication using Blockchain.

NA

Author Response

Greetings,

Kindly find the attached file. We have re=written many sentences and added few subsections and tables/figures to bring more clarity to the document. All of this has been detailed in the response.

Thank you for your comments during this first round of review.

William George

Reviewer 2 Report

A few suggestions to improve the article:

in figures, where possible, increase font sizes because small sizes are difficult to read

figure 4 - in my opinion, this is a table with the corresponding renumbering of figures and tables in the text below

figure 6 - in my opinion, is superfluous and does not bring anything new, and can be excluded

in general, in my opinion, the authors somewhat overuse color pictures, reminiscent of illustrative magazines, and therefore, in general, the article looks less scientific, similar to advertising and educational material

Some illustrative drawings seem quite well-known and trivial, so it is necessary to explain their necessity, as well as indicate either your own authorship or the source of the illustration (including copyrights for mini-images or icons used in the drawings)

Author Response

Greetings

Thank you for your valuable comments. Please find attached our response to your constructive advices. We have re-written the abstract to be more holistic and we have removed an image (as was advised) and added a table (TABLE 3) to position our research in the entire genre.

Thank you for your support during the first round of review of our paper.

Regards
William George

Reviewer 3 Report

The topic is interesting. The problems of implementing blockchain technologies are relevant in supply chain management. Especially for food supply chains.

My comments on the manuscript are as follows:

1. The Abstract must be improved. The major problem in the Abstract is that the objectives and methods of your research are not clearly presented.

2. The relevance of the study is poorly shown. There is no statistical data confirming the relevance. Please add details.

3. The aim of the paper should be clearly indicated in the Introduction.

4. One paragraph should be added to the end of the introduction section, clarifying the structure of the paper.

5. There are no references in section 1 (Lines 46-93). Please add references.

6. To strengthen your literature review and theoretical implications, you need to include more recent and relevant references published in recent years. Please:

- add references to sections 2.1.1-2.1.5 (Lines 141-199).

- add references for 2021-2023 in table 1.

7. The authors presented analysis up to 2019 (Lines 226-244). What changes occurred in 2020-2023? What impact has the Covid pandemic had on food supply chains and blockchain technology? Please explain for the readers.

8. The research methodology (research steps and methods in Section 3) should be provided.

9. The "Discussion" section involves comparing the results obtained by the authors with the results of other studies in this field. There is no such comparison in the article.

10. The scientific novelty of methodology requires a clearer understanding. The authors need to highlight the contribution of the paper.

All the best

Author Response

Thank you for your valuable comments. We have edited the document according to your advices and review. Thank you once again for your support during the first round of our review. 

Looking forward to clarifying any issues.

Regards
William George

Round 2

Reviewer 3 Report

The authors took into account the comments of the reviewers and changed the title and content of the manuscript, added a literature review.

No response was received to comments 2 and 5.